# Estimation of Fugl–Meyer Assessment Upper-Extremity Sub-Scores Using a Mixup-Augmented LSTM Autoencoder and Wearable Sensor Data

**DOI:** 10.3390/s25216663

**Published:** 2025-11-01

**Authors:** Minghao Liu, Hsuan-Yu Lu, Shuk-Fan Tong, Dezhi Liang, Haoyuan Sun, Tian Xing, Xiangqian Shi, Hongliu Yu, Raymond Kai-Yu Tong

**Affiliations:** 1Department of Biomedical Engineering, The Chinese University of Hong Kong, Hong Kong SAR, China; 1155254855@link.cuhk.edu.hk (M.L.); fannytong@cuhk.edu.hk (S.-F.T.);; 2Faculty of Applied Sciences, Macao Polytechnic University, Macao SAR, China; hylu@mpu.edu.mo; 3Institute of Intelligent Rehabilitation Engineering, University of Shanghai for Science and Technology, Shanghai 200093, China; 4Shanghai Engineering Research Center of Assistive Devices, Shanghai 200093, China; 5Key Laboratory of Neural-Functional Information and Rehabilitation Engineering of the Ministry of Civil Affairs, Shanghai 200093, China

**Keywords:** stroke, Fugl–Meyer assessment, IMU, deep learning, remote assessment

## Abstract

**Highlights:**

**What are the main findings?**
A targeted 7-motion protocol enables rapid (10-minute) estimation of FMA-UE subdomain scores—hand, wrist, elbow-shoulder, and coordination—by capturing distinct joint synergies often missed by total-score estimators.The integration of mixup augmentation with an LSTM-based autoencoder attains high generalization (R^2^ > 0.82, Pearson’s correlation coefficient r > 0.90) through leave-one-subject-out validation.

**What is the implication of the main finding?**
A comprehensive motion set is essential for precise estimation across all FMA-UE functional domains, as confirmed by reduced motion analysis.The proposed model offers a clinically viable, objective screening tool for stroke assessment and therapy triage, completable within 10 minutes.

**Abstract:**

Stroke is a leading cause of long-term disability worldwide, necessitating efficient motor function assessment to guide personalized rehabilitation. The Fugl–Meyer Assessment for the Upper Extremity (FMA-UE) is a clinical gold-standard tool, but it is time consuming and requires trained clinicians, which limits its frequency of use and accessibility. While wearable sensors and deep learning offer promising avenues for remote assessment, accurately estimating detailed sub-scores of specific motor functions remains a significant challenge. This work introduces a deep learning framework for automated estimation of FMA-UE total and subdivision scores. Data was collected from 15 participants using four inertial measurement units (IMUs) positioned on the arm and trunk. Each participant performed seven specialized functional motions designed for comprehensive joint synergy involvement within ten minutes. A therapist-rated FMA-UE provided true scores. The proposed model leverages the integration of an LSTM-based autoencoder and mixup augmentation to enhance generalization and robustness. Evaluated through a leave-one-subject-out cross-validation (LOSOCV), the estimator demonstrated strong performance, achieving R^2^ values exceeding 0.82. Pearson’s correlation coefficient r was more than 0.90, and the normalized root-mean-square errors (NRMSE) were below 0.14 for all subparts (A–D). Crucially, the total FMA-UE score was estimated with an NRMSE of 0.0678. These results show that a concise, sensor-based assessment can reliably predict detailed motor function scores.

## 1. Introduction

Stroke is a leading cause of mortality and long-term disability worldwide [1]. In China, the prevalence of stroke has risen significantly over the past decade [2], accounting for nearly one-third of all stroke-related deaths globally [3]. Hemiplegia is one of the most common impairments observed in stroke survivors. In particular, upper extremity (UE) hemiplegia has been shown to significantly affect patients’ performance in activities of daily living (ADLs) and their health-related quality of life (QOL) after stroke [4,5]. Therefore, long-term stroke rehabilitation serves as an essential strategy to help stroke survivors restore motor function and maximize the potential for recovery [6]. In rehabilitation procedures, motor function assessment plays a crucial role by enabling the provision of individualized rehabilitation interventions. [7]. To assess professional guidance, stroke survivors have to attend rehabilitation institutions, such as hospitals and outpatient departments. However, this often includes substantial costs, including temporal, spatial, manpower, and financial resources [8,9], which can cause barriers to accessible and continuous rehabilitation. Many stroke survivors face mobility challenges that limit their ability to access specialists in person. The reason for unavailability stems from multiple aspects. A study reported that nearly half of patients with stroke in the United States cannot access occupational therapy [10]. Therefore, the remote accessibility to standard rehabilitation assessments is expected to be pressingly developed.

The Fugl–Meyer Assessment (FMA) [11] is an assessment tool that is widely accepted and used by therapists to evaluate the motor performance and movement quality of post-stroke individuals, and has been considered as one of the most comprehensive and reliable quantitative measurements of motor impairment [12,13,14,15]. Although possessing excellent reliability and validity, the standard administration of the full FMA typically requires approximately 30 min or even longer to complete [11,14] and must be conducted under the supervision of trained therapists. The FMA-UE, the upper extremity portion of the FMA, still includes 23 items, involving shoulder and elbow, wrist, hand, coordination functional assessment, and remains a time-consuming and labor-intensive process to conduct [14]. Moreover, the evaluation result of these conventional assessment methods more depends on clinical experience, inevitably introducing a degree of subjectivity [16,17]. Furthermore, another limitation of conventional assessment methods is the lack of quantitative and automatic analysis in FMA-UE. To address these issues, there is a growing need to optimize motion set and maximize measurement range in the assessment at the same time, as well as developing quantitative and automated assessment systems for post-stroke assessment.

Modern wearable technology offers the advantages of portable, real-time, noninvasive, and continuous monitoring, and enables personalization and remote possibilities, which have been applied in stroke assessment [18,19,20,21]. The wearable devices based on inertial measurement units (IMUs) have a broad application in motion monitoring and clinical research [22,23,24,25]. Some learning-based methods are utilized in joint motion analysis by using extracted features such as mean, standard deviation, amplitude, and root mean square [26,27,28]. Based on these, some research on predicting FMA scores by IMUs has been published [24,29,30], which enables remote UE function assessment [31,32], thus facilitating personalized rehabilitation intervention [33]. Learning-based methods are usually implemented based on machine learning and deep learning; the most commonly used algorithms include support vector machine (SVM), random forest (RF), artificial neural networks (ANNs), deep neural networks (DNNs), convolutional neural networks (CNNs), and long short-term memory (LSTM) [34,35,36]. Due to the advantages of LSTM in processing time-series signals, it has been used to more accurately process data in stroke impairment assessment predictions [37,38], exhibiting potential in remote evaluation systems.

Existing methods for estimating FMA-UE scores typically rely on motion data collected from either the entire, partial, or summarized sets of movements specified in the FMA-UE assessment items [24,30,39,40,41,42,43]. Specifically, Formstone et al. combined IMUs, mammography (MMG), and machine learning algorithms to predict FMA-UE scores [24]; this multi-sensing method provides multidimensional signals and fine motor detection. However, it may increase the amount of data and inter-subject variability, which should be considered. Adans-Dester et al. [28] used eight motions of the Wolf Motor Function Test to estimate the FMA score based on the Functional Ability Scale (FAS), simplifying the traditional standardized assessments. Some work developed an estimator that can predict FMA-UE scores by using wearable IMU data from four activities of daily living (ADLs) and point-to-point motions [44,45] or three volitional reaching motions [30]. These studies substantially reduce the number of motions required from subjects compared to traditional assessment procedures. Another study used 1–2 min of random arm movements as the input for FMA prediction [29], optimizing the assessment procedure regarding time dimensions. However, the learning-based methods mentioned above mainly focus on the total FMA or general FMA-UE motor function scores; there are no detailed subdivision scores that can evaluate corresponding joint motor functions, such as wrist, elbow, and shoulder. Two subjects with identical total FMA-UE scores may suffer from various functional disabilities, and only by making detailed sub-score predictions can these essential differences be revealed. In FMA-UE, the whole motor function assessment is divided into four parts according to the joints in major consideration: (1) part A: shoulder and elbow; (2) part B: wrist; (3) part C: hand; and (4) part D: coordination and speed [11]. With these sub-scores of FMA-UE, the functional disorders would be evaluated, enabling therapists to develop highly personalized rehabilitation plans, accurately assessing the effect of intervention measures and track recovery trajectory, which potentially offer precision and individuation to remote stroke assessment.

From a technical perspective, the extensive scope ensures that a wide spectrum of upper limb movement characteristics is captured, which in principle contributes to high assessment accuracy. However, this comprehensiveness comes with a cost, as adopting all upper limb motion features for automated analysis substantially increases computational complexity. These inherent limitations—elevated temporal, manpower, and computational costs—hinder its practicality for frequent administration, remote monitoring, and quick feedback. In this work, an estimator was developed to predict FMA-UE sub-scores using wearable IMU data of seven motions. The seven motions we selected were summarized from FMA-UE, based on incorporating basic joint movements across the frontal, transverse, and sagittal planes or exposing a fundamental synergistic pattern and multidimensional control capabilities, which related to specific joint motor functions: (1) reaching upward (RU); (2) reaching contralateral knee to ipsilateral ear (RKE); (3) hand to lumbar spine (HTS); (4) elbow pronation–supination (EPS); (5) wrist circumduction (WC); (6) hand mass flexion and extension (HMFE); and (7) reaching knee to nose (RKN). We aim to connect the characteristics of specialized motions to the FMA-UE-corresponding part A (shoulder and elbow), part B (wrist), part C (hand), and part D (coordination and speed). The estimator was trained from a dataset of fifteen subjects (four healthy adults and eleven stroke survivors) with four IMUs on trunks and arms for capturing a large set of motion features and compensatory strategies [46]. Subjects were instructed to finish FMA-UE assessment with only one extra upward reach because most specialized motions are included in FMA-UE. The motions were scored in real-time by therapists during the assessment. The seven specialized motions (as input data) and FMA-UE sub-scores (as criteria) were used in estimator training. An autoencoder was applied for auto feature selection in the LSTM-based network, and a regression model was evaluated by leave-one-subject-out cross-validation (LOSOCV). The determination coefficient (R^2^), Pearson’s correlation coefficient (r), the mean-average error (MAE), normalized MAE (NMAE), root-mean-square error (RMSE), and normalized RMSE (NRMSE) were also analyzed. To validate the importance of the full motion set, estimators trained by reduced motion sets were also evaluated, and the results showed that the full motion set achieved the lowest NMAE and NRMSE across all evaluations. Compared to existing works, our method distinguishes the difference of functional motion disabilities between subjects who have similar FMA-UE general scores, which are reflected in the FMA-UE subdivision scores, offering an automated, precise, personalized, and optimized solution in remote stroke assessment scenarios.

## 2. Materials and Methods

### 2.1. Data Collection Method

As a part of the data collection, we first enrolled the suitable participants, including heathy subjects and stroke survivors. In the experiment period, participants were expected to wear IMU sensors and perform FMA-UE under the supervision of one skilled therapist. The IMU data were collected by the system we developed.

#### 2.1.1. Participants

Fifteen subjects were enrolled in the FMA-UE experimental protocol (Table 1). It was registered on ClnicalTrials.gov (ID: NCT03286309). All study protocols adhered to the principles of the Declaration of Helsinki. The Joint Chinese University of Hong Kong–New Territories East Cluster granted ethical approval for the study. Specifically, four healthy adults and eleven patients with stroke were included (age: 50.93 ± 17.31 y/o, ranging from 26 to 74 years old; FMA-UE total score: 49.80 ± 17.05, ranging from 14 to 66; in subdivisions, part A scored 28.80 ± 8.80 from 4 to 24; part B scored 6.73 ± 4.09 from 0 to 10; part C scored 10.33 ± 4.40 from 0 to 14; and part D scored 3.93 ± 1.75 from 1 to 6). The stroke participants were recruited for the study from local rehabilitation centers and communities in Hong Kong. The following inclusion criteria were applied: (1) in chronic stage (6 months after the onset of stroke) with a pure unilateral motor paresis after a stroke, (2) sufficient cognition to follow simple instructions and understand the content and purpose of the study, (3) followed FMA-UE experimental protocol. The data that included all FMA-UE items were used for analysis.

The exclusion criteria for stroke participants were as follows: (1) participants with severe dysphasia with inadequate communication, (2) any additional medical or psychological condition affecting their ability to comply with the study protocol, and (3) history of other neurological disease psychiatric disorders, including alcoholism and substance abuse. Before the start of the experiments, written informed consent was obtained from each participant, following the protocol approved by the local institutional review board.

#### 2.1.2. Experimental Procedures

Four IMU sensors (Hi221, Sea Land Technologies Co., Ltd., Taiwan, China) were switched on in a stable state for 3 s to make reliable measurement, because a self-calibration was conducted in Hi221 when the power was on. For the first use of the IMUs, the calibration program took an average 10 seconds to remove any offset, which was a standard IMU processing technique to prevent drift due to sensor bias. The subject stayed in a neutral position with their paretic arm, with all IMUs aligned. The initial orientation of the IMUs was computed from this stationary neutral position [47]. All the IMUs were firmly and precisely attached to four segments (hand, forearm, upper arm, and trunk as shown in Figure 1) [30,48] by using double-sided tape to avoid potential sensor displacement during long-duration or repeated movements. Thus, all the IMUs were initialized with the same target heading angle [49]. They were placed along with positive z axes aligning with the anteroposterior axis and pointing to the body, and the x, y axes aligned with the transverse, longitudinal axis of the arm or trunk.

After donning, the participants needed to follow instructions from the licensed therapist to finish all the FMA-UE items. Any physical support from the unaffected side was not allowed during the assessment. We started to record IMU data when the participants got ready to perform seven specialized motions: (1) RU, (2) RKE, (3) HTS, (4) PS, (5) WC, (6) HMFE, and (7) RKN. All motions needed to be repeated about five times. The IMUs were able to reflect the rotational state of the body in three-dimensional space and are commonly used to describe the range of motion and angles of joints and tilting. We stopped recording and saved the IMU data when completing each motion. The therapist scored part A, B, C, or D of the FMA-UE, which were considered as training criteria after finishing each subdivision. Only one extra motion was needed due to most of them having been contained by FMA-UE.

#### 2.1.3. Hardware and Software

The whole collection system consisted of wearable devices and a computer. The IMUs were constructed via the HI221 chip. The IMUs’ accelerometer featured a measurement range of ±8 g. The gyroscope offered a range of ±2000°/s with a size of 20 × 38 × 8.5 mm (W × L × H), with an average working duration of 8 h after charging. An affiliated dongle (Sea Land Technologies Co., Ltd., Taiwan, China) for the IMUs was used to collect the participants’ motion signals (acceleration, gyroscope, and orientation on the three axes) in real time from a board filter on the HI221. After plugging the dongle into a computer, the IMU data were wirelessly transmitted to the dongle. A computer client was also developed by Python (version 3.12.1) to receive dongle data via serial communication, plot real-time data, and save data on the computer. In addition, the joint angles (wrist, elbow, shoulder) on the three axes were calculated on the computer client. The sampling frequency of the collection system was set to 100 Hz.

#### 2.1.4. Specialized Upper Limb Motions from FMA-UE

The seven selected upper limb motions were summarized from FMA-UE according to its subdivisions, part A, B, C, and D, which reflected the motor functions of shoulder and elbow, wrist, hand, and coordination and speed, respectively. These motions were intended to be used to estimate FMA-UE sub-scores so that we could infer different motor functions.

For each motion, the participants needed to fully extend the elbow and reach as high as they can when performing RU, which was mainly contributed by shoulder abduction on the frontal plane, shoulder flexion on the sagittal plane, shoulder internal/external rotation, and elbow extension/flexion. When participants were asked to perform RKE, reach the contralateral knee against the impaired side to the ipsilateral ear, they were expected to more focus on shoulder flexion/extension on the transverse plane and shoulder extension on the sagittal plane; the elbow extension/flexion and elbow pronation/supination were also included. As for the shoulder extension on the sagittal plane, we selected HTS, in which participants were requested to touch the back lumbar spine by hand. However, the elbow pronation/supination evaluation in mixing synergies may have been affected by potential compensation from the shoulder due to impaired synergy ability, so we added EPS. Participants performed EPS at a neutral position (shoulder at zero degree) and the elbow at ninety degrees to isolate potential compensation from the shoulder. The four motions, RU, RKE, HTS, and EPS, were used to estimate shoulder and elbow (part A of FMA-UE), which we marked as A*.

For wrist, we concluded the wrist motion as WC, including repeat, stability, and range characteristics. Participants were instructed to complete five repetitions of wrist circumduction in a stable manner with a ninety-degree elbow and achieve the full active range of motion during movement. Individuals with motor impairments often rely on compensatory movements from proximal joints—such as the wrist, elbow, and trunk—to complete hand-related tasks. To account for this, we designed the HMFE task to specifically evaluate full flexion and extension, ensuring these compensatory patterns are captured during assessment. Finally, a coordination motion was chosen, RKN, as it emphasized upper limb multi-joint coordination, which depended on the proprioception, a sense of the brain to always know the position of each part of the body in space. We marked WC, HMFE, and RKN as B*, C*, and D* because these were intended to be the estimators of wrist, hand, and coordination (part B, C, and D of FMA-UE). As detailed in Table 2, the proposed set of seven specialized motions comprehensively encompassed the functional scope of all 23 original FMA-UE motion items. The table explicitly maps each selected motion to its corresponding FMA-UE assessment item, demonstrating complete coverage of the clinical evaluation.

### 2.2. AI Model Training and Evaluation

As part of the model training and evaluation method, we first applied some processing methods to the data, including data preprocessing and feature generation. The LSTM-based estimator was built by a feature selection layer and prediction layer, and was trained by using generated features and criteria scores. Finally, the models were evaluated through LOSOCV.

#### 2.2.1. Data Processing

Based on raw data, the relative angles between the IMUs of each arm segment were also computed, such as shoulder elevation angles, elbow flexion or extension angles, and wrist flexion or extension angles (Figure 2a). Then, we applied consistent preprocessing methods to the raw data and joint angle: (1) we removed the outlier and replaced it with the mean value from the selected samples, (2) a bandpass Butterworth filter with a cutoff frequency of 10 and 50 Hz was applied to eliminate power frequency interference [30,50], (3) resampling to 100 Hz was performed to ensure a consistent sample rate. As for data augmentation, we sequentially applied the following methods for improving robustness: (1) Gaussian noise, (2) time warping by stretching or compressing the time scale, (3) random amplitude scaling by stretching or compressing the amplitude factor (0.8~1.2) [30].

After augmentation, we used sliding window for the series data, which is commonly used for time-series data processing [51]. Here, we windowed each motion into 2 s series with overlaps of 50%. In order to further format the data as training input, we generated features of all data windows. For each window, a set of features was generated by feature generation function based on the IMU data and joint angles. The feature generation function mainly included mean, standard deviation (std), root mean square (rms), amplitude, maximum and minimum, approximate entropy, and range of signal [27]. For each IMU datum, we further generated features of acceleration, gyroscope, orientation acceleration, angular difference, and inclination ratio (based on the trunk). We generated 1530 features in total; these calculated features were capture signal characteristics, considering patterns related to motion of each participant.

#### 2.2.2. AI Model Design

To format the model input as Figure 2b shows, the features of seven motions were combined into a vector, and the normalization of features and true scores was essential for avoiding value loss and optimizing training. For each motion input, it had a feature vector (tensor) shaped in [windows length (WL), 1530]. After processing, the WL was used for the sliding window setting. There were seven motions for the model input, then the seven feature vectors were concatenated into a tensor [7, windows length, 1530] as the input, which are shown in Figure 2c.

For better generalization and a more robust estimator, a mixup method was selectively applied to the input before the LSTM network to enhance generalization while preserving the underlying real-data distribution [52,53]. It is an augmentation method that had been proven to significantly improve outcomes in deep learning fields [54,55,56,57,58,59,60]. Here, mixup generated a new sample datum for model input and new true scores for loss function by proportionally adding two data samples according to the following formula:(1)$$\hat{x}\;=\;\lambda x_{i}\;+\;(1\;-\;\lambda )x_{j},$$(2)$$\hat{y}=\lambda y_{i}+\left(1-\lambda \right)y_{j},$$(3)λ ~ Beta(α , α),
where $x_{i}$, $x_{j}$, are two feature vectors, $y_{i}$, $y_{j}$ are two true score vectors. They were drawn randomly from the dataset, and λ follows a Beta distribution with parameters α, $\lambda \;\in \left[0,\;1\right]$. The α was set to 1.

After mixup, an LSTM-based auto feature selection was used to encode features, reducing the dimensionality of the feature space and automatically extracting information from the large size features for predicted FMA-UE sub-scores. In the auto-encoder phase (Figure 2c), the input, characterized by a shape of [7, window length, 1530], was first processed by an LSTM layer to capture temporal patterns and feature dependencies. This step resulted in a 256-dimensional hidden state at each time step. An average pooling layer was applied to the sequence across all WL time steps, condensing the temporal information for each motion into a single 256-dimensional vector. Finally, the generated [7, 256] vector was subsequently passed through a linear layer, which projected each 256-dimensional motion representation down to a compressed 128-dimensional feature vector, resulting in the final encoded feature vector of shape [7, 128], which is a latent feature [61,62].

In the score prediction period, we chose one LSTM layer, and one linear layer to be the predictor by using the selected features of LSTM had been shown to be suited to deal with tasks involving sequence data [63]. Here, LSTM was designed to model the interdependencies among the latent features. While processing the sequence, the hidden dimension was maintained. Then, the output vector at the final time step was selected, serving as a comprehensive representation of all the latent features. This representation was then passed through a dropout-regulated linear layer to yield the final output of shape [1, 4], which corresponded to the four FMA-UE sub-scores.

In the training period, the MSE loss function can compute the error between the estimated scores and the mixed up true scores, and model’s weights were updated by the optimization algorithm (Adam was used in this work). According to the error, the encoder and predictor were optimized for optimal feature selection and accurate score prediction. The trained estimators were saved and used for evaluation by LOSOCV in the test period. The hyperparameters were set in this work as follows: epochs = 100; learning rate = 0.00005; input size = 1530; number of recurrent layers = 1; output size = 4; dropout = 0.3.

#### 2.2.3. Model Evaluation

According to LOSOCV, the estimator needed to be trained and tested fifteen times. In each iteration, model hyperparameters were initialized. Data of fourteen subjects were used for training, the data of the left-out subject were used for testing, and the data of training set and test set were separated. The tested scores of FMA-UE subdivisions were recorded. To evaluate estimator performance, R^2^, MAE, NMAE, RMSE, and NRMSE of sub- and overall scores were calculated for estimator evaluation after fifteen iterations. The R^2^ was used to determine the fitting goodness, representing the ability of the model to explain variance. For estimation accuracy, MAE can explain the absolute average gap between the estimated scores and the true scores, while RMSE measures the “standard deviation” of the difference between the estimated scores and the true scores. Moreover, we normalized estimated scores and the true scores due to their different scales in order to evaluate overall performance of the estimator. Furthermore, models of different motion sets and motion combinations were compared.

## 3. Results

### 3.1. Accuracy of Estimator

The trained FMA-UE sub-score estimator was evaluated by using the mean estimates through LOSOCV. As shown in Figure 3a–d, which plot-estimated scores against true scores for each FMA-UE subdivision, the model demonstrated strong predictive performance across all parts: Part A achieved an R^2^ of 0.8776, a correlation coefficient r of 0.9368, an NMAE of 0.0751 (MAE = 2.7054), and an NRMSE of 0.0998 (RMSE = 3.1927). Part B attained an R^2^ of 0.9151, a correlation coefficient r of 0.9566, an NMAE of 0.1202 (MAE = 1.2023), and an NRMSE of 0.1316 (RMSE = 1.3160). Part C yielded an R^2^ of 0.8264, a correlation coefficient r of 0.9090, an NMAE of 0.0974 (MAE = 1.3633), and an NRMSE of 0.1358 (RMSE = 1.9007). Part D reached an R^2^ of 0.9077, a correlation coefficient r of 0.9527, an NMAE of 0.0787 (MAE = 0.4722), and an NRMSE of 0.1257 (RMSE = 0.5597). These results are summarized in Table 3. As shown in Table 4, the absence of mixup augmentation led to a modest reduction in accuracy for part A. In contrast, more substantial performance degradation was observed for parts B, C, and D, marked by notable decreases in R^2^ and correlation coefficient (r), alongside increased error metrics (NMAE and NRMSE).

For the comprehensive evaluation, all estimated scores and true scores were analyzed. Figure 3e shows an overall fitting curve based on all estimated scores and true scores with 0.9710 for R^2^ and 0.9854 for correlation coefficient r. Part A, B, C, and D of FMA-UE have different scoring scales. Specifically, part A ranges from 0 to 36, part B from 0 to 10, part C from 0 to 14, and part D from 0 to 6. The overall accuracy was calculated from weighted scores, with 0.0929 for NMAE and 0.1144 for NRMSE (Table 5). Moreover, in order to validate the performance of the total score, total FMA-UE scores were computed by adding sub-scores, which are shown in Figure 3f. The estimator achieved 0.9514 for R^2^, 0.9753 for correlation coefficient r, 0.0484 for NMAE, and 0.0678 for NRMSE, confirming its reliability and precision for comprehensive motor function assessment.

### 3.2. Comparisons with Different Reduced Motion Set

To understand the importance of each motion set, we trained the estimator by using several reduced motion sets, then the accuracy of these estimators was compared with the estimator that was trained by using the whole motion set. All evaluations of reduced motion adopted LOSOCV as the same mentioned above. As Figure 4a,b show, the estimator from the reduced motion set was compared in accuracy to the specific part that related to reduced motion, which reached about two times that of NMAE and NRSME based on the estimator from the full motion set. The overall evaluations of the estimator from the reduced motion set are shown in Table 5. In the B*C*D* motion set, R^2^, correlation coefficient r, NMAE, and NRMSE could reach 0.8767, 0.9364, 0.1896, and 0.2330, respectively. In the A*C*D* motion set, R^2^, correlation coefficient r, NMAE, and NRMSE achieved 0.9482, 0.9737, 0.1326, and 0.1778, respectively. In the C*-removed set, the accuracy reached 0.9112 in R^2^, 0.9546 in correlation coefficient r, 0.1437 in NMAE, and 0.2011 in NRMSE. Finally, we removed the D* set; the estimator only achieved 0.9150 for R^2^, 0.9565 for correlation coefficient r, 0.1553 for NMAE, and 0.1988 for NRMSE. In addition, to quantify the importance of each motion in A*, estimators from reduced A* motion were also evaluated. As shown in Figure 4c, the use of the full A* motion resulted in the lowest NMAE and NRMSE values, indicating that the comprehensive inclusion of all motions is essential for maximizing estimation precision and ensuring robust performance for all subdivisions of the FMA-UE.

## 4. Discussion

In this work, we developed an FMA-UE sub-score estimator trained by using wearable-IMU data. The data collected from seven specialized upper limb motions were summarized from FMA-UE, aiming to estimate specific motor functions and coordination related to subdivisions of FMA-UE. Compared to existing FMA-UE estimators, our method was able to conduct a detailed assessment requiring seven simple motions and distinguished variations in functional motion impairments rather than FMA-UE general scores, offering an automated, precise, personalized, and optimized solution for future stroke assessment.

Because the motor function of joints is affected by stroke [64,65,66], the selection of seven upper limb motions is critically important for providing a comprehensive and accurate estimation of motor function as defined by the FMA-UE subdivisions (part A, part B, part C, and part D). The motions RU, RKE, HTS, and EPS (A*) were strategically chosen to collectively assess the synergy motor functions of the shoulder and elbow. Specifically, they target shoulder movement planes (sagittal, frontal, transverse) or isolate individual joints, such as EPS, to evaluate elbow pronation/supination without compensation from shoulder movements, thus ensuring a reliable assessment of part A. For the wrist, the WC (B*) motion is essential as it directly evaluates crucial aspects of wrist motor function: repeatability, stability, and range of motion. The HMFE (C*) motion is vital for assessing hand function by hand mass full flexion and extension. Its importance lies in that it was designed to detect potential compensatory movements or tremors, which are common in individuals with poor motor control. Finally, the RKN (D*) motion is key for evaluating coordination and speed; its importance mainly stems from multi-joint coordination and proprioception.

The distribution of seven specialized motions, as visualized in Figure 5, revealed that our optimized motions do not function as a monolithic block but rather as a set of complementary tasks that differentially probe distinct aspects of upper-limb motor function. Crucially, the model learned to weigh these motions in a manner that aligns closely with established clinical knowledge. Specifically, the robust contribution of RU and RKE to part A (shoulder and elbow) was expected, as these complex, multi-joint tasks inherently challenge the range of motion, strength, and motor control of the upper limb. Similarly, the strong association of WC with part B (wrist) validated its selection as a core task for capturing the compound mobility and stability of the wrist joint. The model did not rely on a single task but instead synthesized information from multiple motions. This suggested that our model can capture properties of movement that emerge across different motor challenges. This differential contribution map highlighted a key advantage of our approach: it moved beyond an FMA-UE total score to offer a functional profile of a patient’s motor impairment. For instance, two patients with an identical total FMA-UE score may exhibit profoundly different contribution patterns: one with deficits primarily in proximal tasks (low in RU or RKE contribution) and another in distal coordination (low in EPS or WC contribution). Such a profile could ultimately guide more targeted, personalized rehabilitation strategies by pinpointing the specific functional domains that require intervention.

To quantitatively evaluate the effectiveness of the method proposed in this study, we conducted a comprehensive comparison against the recently published FMA-UE prediction work (Table 6). Although these comparative studies differ in sensor configuration, task sets, and the models used, our method demonstrates compelling advantages in prediction performance. Specifically, the LSTM estimator in this study achieved the best performance in predicting the total score of FMA-UE (R^2^ = 0.9514, NRMSE = 0.0678), surpassing other multi-IMU studies such as Adams-Dester et al. (R^2^ = 0.86) [28] and Oubre et al. (R^2^ = 0.75) [29,44]. In addition, our model achieved the lowest NRMSE (0.0678) among all comparable studies, demonstrating superior prediction accuracy even when compared to the approach by Zhou et al. [30], which utilized only three actions (NRMSE = 0.0698). Particularly noteworthy is that our method is the approach that can provide highly accurate predictions for the four sub-scores of part A (shoulder/elbow), part B (wrist), part C (hand), and part D (coordination) with R^2^ values all above 0.82. This demonstrates that our method based on seven specialized motions can comprehensively and precisely capture the movement characteristics of the upper limb, offering a precise, personalized, and optimized solution in remote stroke assessment scenarios with significant clinical significance. In terms of task design, our protocol requires seven specialized motions, making it far more efficient than administering the full 33-item FMA-UE. More importantly, it strikes an effective balance between brevity and comprehensiveness: while studies using fewer or more generic actions (e.g., 1–2 min of routine motion or 3–4 ADLs [29]) achieve speed, our more representative and targeted task set yields significantly superior prediction performance without compromising temporal efficiency.

The extensive pool of motion-related features was generated in preprocessing. Then, these features were subsequently processed by an LSTM-based autoencoder, which shows critical functions: it selected the most informative features and reduced the dimensionality of the feature space based on their contribution to outcome score prediction. A key strength of this approach lies in its capability to effectively extract and leverage movement information from a large pool of motion features, enabling robust motor function assessment. To further enhance generalization and robustness, the mixup augmentation technique was applied to the input features prior to the LSTM network. This method artificially expands the training set by creating new synthetic samples and corresponding labels through linear coalescing between two randomly selected data points, as defined in Equations (1)–(3). By doing so, mixup encourages the model to learn smoother decision boundaries and reduces overfitting, thereby significantly improving the ability to generalize unseen data. In Table 3 and Table 4 and with the absence of mixup, the predictive performance of part B and part D is relatively weaker than that of part A and C (with lower R^2^ values and greater error). By using mixup, the estimator achieved R^2^ values higher than 0.82 and the correlation coefficient r reached more than 0.90. In particular, the improvement in part B is the most remarkable, such as the R^2^ jumping from 0.6474 to 0.9151 and the R^2^ of part D also significantly increasing from 0.7988 to 0.9077. These prove that mixup effectively addresses the core issue of insufficient generalization and unreliability due to small sample size. The coalescence helps mitigate the risk of model overfitting and enhances feature encoding, contributing to a more reliable and stable prediction performance across diverse patient profiles.

Based on the current work, several limited aspects should be presented for future improvement. First, although the mixup and LOSOCV were applied against small sample size, the amount and diversity of the dataset could be significantly enhanced by enrolling more participants, particularly those with severe (for instance, FMA-UE scores less than 10) and very mild impairments to improve model generalizability across all impairment levels. Furthermore, non-paretic sides were not evaluated in this work; expanding data collection of non-paretic sides would provide a more comprehensive movement dataset. In data collection, although consistency in sensor placement was carefully pursued, variations remain a challenge. Future studies could consider more rigorous standardization protocols or explore multiple sensor techniques to improve measurement accuracy. As for the seven specialized motions, this study only investigated the impact of reduced known motion sets on estimation accuracy; future work should refine and optimize the selection of functional motions aligning with FMA-UE subdivisions and include more synergistic relationships to enhance both clinical relevance and estimator performance.

The proposed FMA-UE estimator, capable of predicting four sub-scores (parts A–D), holds significant potential for integration into remote rehabilitation platforms. By providing objective feedback on shoulder/elbow, wrist, hand, and coordination functions, the system enables therapists to precisely monitor patient progress between clinical visits and tailor home-based exercise regimens accordingly. This continuous, data-driven assessment not only facilitates personalized rehabilitation but also demonstrates the long-term clinical value of the technology in supporting scalable, tele-rehabilitation services and improving functional outcomes in real-world settings.

Lastly, the current evaluation focused primarily on motor function involving the shoulder, elbow, wrist, hand, and coordination. Expanding the system to include additional sensors, such as integrating force sensors for reflexes and fine hand function assessment, could offer a more complete implementation of the FMA-UE and potentially create a comprehensive functional evaluation.

## 5. Conclusions

In conclusion, we developed an FMA-UE sub-score estimator based on four IMUs that can assess upper-limb motor functions by performing seven specialized motions, providing more detailed subdivision score predictions compared to general FMA-UE estimators. The seven specialized motions were selected because they involve comprehensive joint motor functions and are related to the subdivisions of the FMA-UE. LSTM networks were employed for the estimator, and the autoencoder and mixup were embedded to enhance generalization and robustness, thus enabling acceptable estimations through the LOSOCV, with an acceptable predictive performance with a true score that was provided by therapists for all subdivisions of the FMA-UE. Moreover, the study of reduced motion sets was validated for demonstrating the importance of comprehensive inclusion of all motions for reliable estimations. These results indicate that combining wearable sensors with deep learning algorithms has great potential to assess FMA-UE subdivisions in stroke survivors, offering the development of personalized rehabilitation programs, and accurately assessing the effect of intervention measures and recovery trajectory tracking in remote stroke assessment.

## Figures and Tables

**Figure 1 sensors-25-06663-f001:**
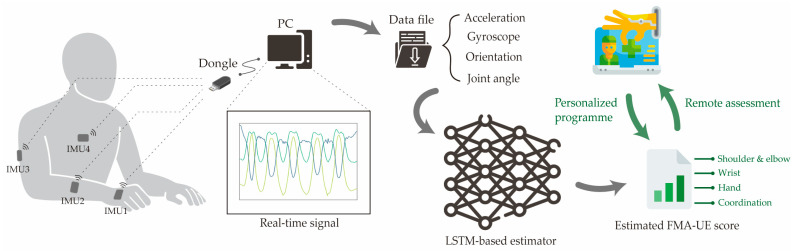
Schematic diagram of the FMA-UE sub-score estimator with four IMUs (IMU1, IMU2, IMU3, IMU4) placed onto paretic hand, forearm, upper arm, and trunk.

**Figure 2 sensors-25-06663-f002:**
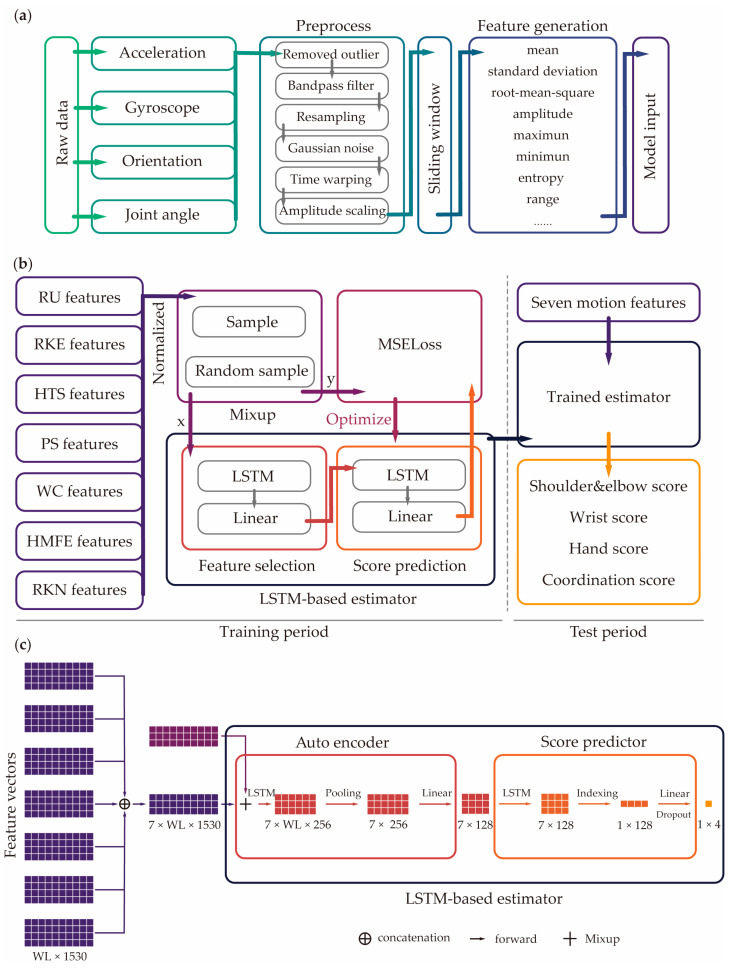
Illustration of data preprocessing and the proposed estimator. (**a**) Data preprocessing, window segment and feature generation from raw data to formatted input. (**b**) The training period and test period procedure. The mixup was utilized for input augmentation (x). The augmented FMA-UE subdivision scores (y) were used in calculating the MSE loss. (**c**) Network architecture of the estimator based on seven specialized motions.

**Figure 3 sensors-25-06663-f003:**
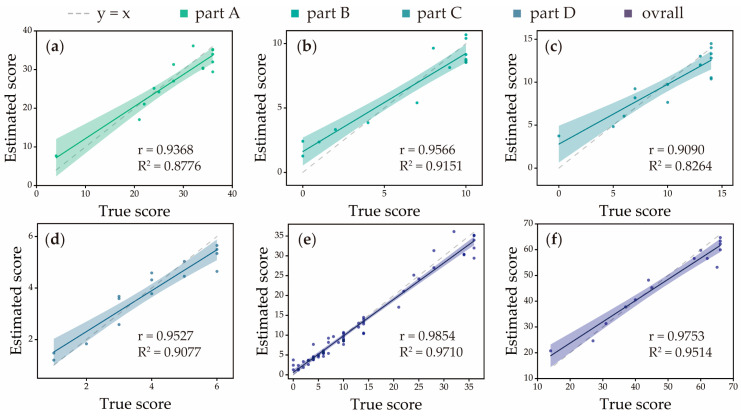
R^2^ and Pearson’s correlation coefficient (r) for part A (**a**), part B (**b**), part C (**c**), and part D (**d**) of FMA-UE scores through LOSOCV. (**e**) The overall evaluation result of the estimator. (**f**) The evaluation result for total FMA-UE score of each participant.

**Figure 4 sensors-25-06663-f004:**
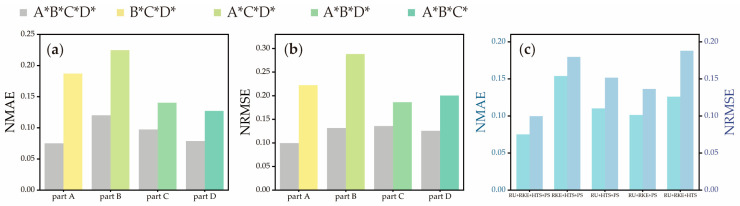
The NMAE (**a**) and NRMSE (**b**) with or without A*, B*, C*, and D*. (**c**) The NMAE and NRMSE of different motion combinations in A*.

**Figure 5 sensors-25-06663-f005:**
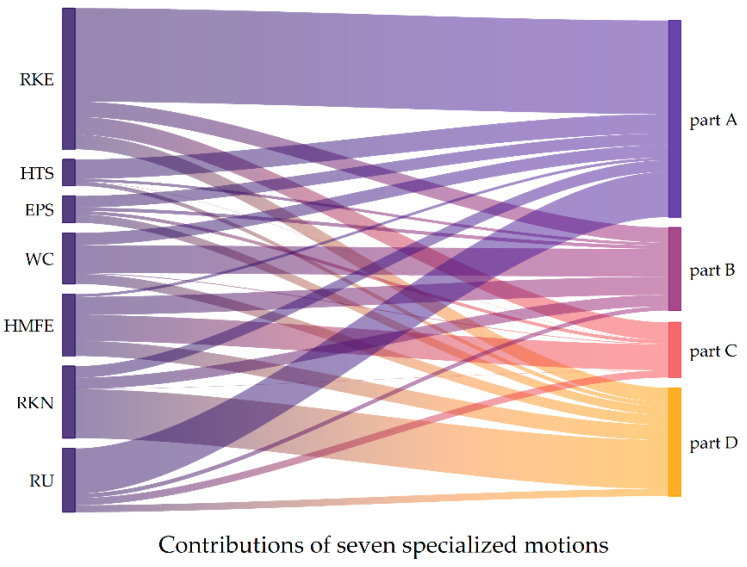
Contribution of each specialized motion to sub-score estimation in this study.

**Table 1 sensors-25-06663-t001:** Demographic and clinical characteristics of participants.

Characteristics	Value	Mean ± Standard Deviation	Min–Max
Age	-	50.93 ± 17.31	26–74
Gender (male/female)	7/8	-	-
Tested side (left/right)	4/11	-	-
FMA-UE score	-	49.80 ± 17.05	14–66
Part A	-	28.80 ± 8.80	4–24
Part B	-	6.73 ± 4.09	0–10
Part C	-	10.33 ± 4.40	0–14
Part D	-	3.93 ± 1.75	1–6

**Table 2 sensors-25-06663-t002:** Description of seven specialized motions (√ indicates that the FMA-UE motion item was involved).

Motion	Description	FMA-UE Motion Items																						
		Part A								Part B					Part C							Part D		
		1	2	3	4	5	6	7	8	9	10	11	12	13	14	15	16	17	18	19	20	21	22	23
Reaching upward (RU)	Paretic limb lifting up				√		√	√																
Reaching knee to ear (RKE)	Hand from contralateral knee to ipsilateral ear, then from ipsilateral ear to contralateral knee	√	√		√		√		√															
Hand to lumbar spine (HTS)	Start with hand on lap			√			√																	
Elbow pronation–supination (EPS)	Elbow at 90 degrees, shoulder at 0 degrees					√			√															
Wrist circumduction (WC)	Elbow at 90 degrees, forearm pronated, shoulder at 0 degrees								√	√	√	√	√	√										
Hand mass flexion and extension (HMFE)	Fully active extension and fully active flexion														√	√	√	√	√	√	√			
Reaching knee to nose (RKN)	Closed eyes, five times	√	√		√																	√	√	√

**Table 3 sensors-25-06663-t003:** The estimator evaluation results for each subdivision of FMA-UE through LOSOCV with mixup (* indicates significant linear correlation between estimated scores and true scores).

Subdivision of FMA-UE [11]	R^2^	r	MAE	NMAE	RMSE	NRMSE
Part A	0.8776	0.9368 *	2.7054	0.0751	3.1927	0.0998
Part B	0.9151	0.9566 *	1.2023	0.1202	1.3160	0.1316
Part C	0.8264	0.9090 *	1.3633	0.0974	1.9007	0.1358
Part D	0.9077	0.9527 *	0.4722	0.0787	0.5597	0.1257

**Table 4 sensors-25-06663-t004:** The estimator evaluation results for each subdivision of FMA-UE through LOSOCV without mixup (* indicates significant linear correlation between estimated scores and true scores).

Subdivision of FMA-UE [11]	R^2^	r	NMAE	NRMSE
Part A	0.8606	0.9330 *	0.0786	0.1056
Part B	0.6474	0.8046 *	0.1823	0.2419
Part C	0.7647	0.8745 *	0.1102	0.1506
Part D	0.7988	0.8937 *	0.1130	0.1913

**Table 5 sensors-25-06663-t005:** The overall evaluation results of the estimator for various motion sets through LOSOCV (A*: motion set A. B*: motion set B. C*: motion set C. D*: motion set D. * indicates significant linear correlation between estimated scores and true scores).

Motion Set	R^2^	r	NMAE	NRMSE
A*B*C*D*	0.9710	0.9854 *	0.0929	0.1144
__B*C*D*	0.8767	0.9364 *	0.1896	0.2330
A*__C*D*	0.9482	0.9737 *	0.1326	0.1778
A*B*__D*	0.9112	0.9546 *	0.1437	0.2011
A*B*C*__	0.9150	0.9565 *	0.1553	0.1988

**Table 6 sensors-25-06663-t006:** Quantitative comparisons with recent published work.

Authors	Sensor	Task Set	Model	FMA-UE Subdivision				Total FMA-UE
				Part A	Part B	Part C	Part D	
Oubre et al. (2020) [29]	1 IMUs	1–2 min motions	Support vector regressor	-	-	-	-	R^2^ = 0.70 NRMSE = 0.182
Adans-Dester et al. (2020) [28]	5 accelerometers	33 items	Modified balanced random forest	-	-	-	-	R^2^ = 0.86
Oubre et al. (2022) [44]	3 IMUs	4 ADL motions	Random forest	-	-	-	-	R^2^ = 0.75 NRMSE = 0.170
Zhou et al. (2025) [30]	4 IMUs	3 motions	Support vector regressor	-	-	-	-	R^2^ = 0.67 NRMSE = 0.069
Present Study (2025)	4 IMUs	7 motions	LSTM	R^2^ = 0.87	R^2^ = 0.91	R^2^ = 0.82	R^2^ = 0.90	R^2^ = 0.95 NRMSE = 0.067

## Data Availability

The data presented in this study are available on request from the author.

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
