# Peer review of "Estimation of Fugl–Meyer Assessment Upper-Extremity Sub-Scores Using a Mixup-Augmented LSTM Autoencoder and Wearable Sensor Data"

_sensors, 2025, doi:10.3390/s25216663_

Round 1
Reviewer 1 Report
Comments and Suggestions for Authors
The manuscript “Estimation of Fugl-Meyer Assessment Upper Extremity Sub-scores using a Mixup-Augmented, LSTM-Autoencoder and Wearable Sensor Data” proposes a framework based on wearable IMU sensors and deep learning models for the automatic assessment of upper limb Fugl-Meyer (FMA-UE) scores in stroke patients. The experimental design considers clinical practicality and adopts a rigorous “leave-one-subject-out cross-validation (LOSOCV)” approach for evaluation. Overall, this is a high-quality and innovative research paper with important clinical application prospects. The revision comments mainly focus on clarification of methods, validation of results, and improvement of the discussion. The authors should respond in detail.
- This study included only 15 participants, including 4 healthy subjects and 11 stroke patients. However, this sample size is relatively small for deep learning models. The authors need to discuss more explicitly the impact of sample size on model stability.
- The article proposes a simplified assessment protocol consisting of 7 specific tasks, which is a core innovation. However, the theoretical rationale and selection process for these tasks are insufficiently described in the introduction and methods. It is recommended to conduct factor analysis or similar statistical validation to demonstrate that these 7 tasks indeed represent the core motor functions of the FMA-UE sub-scores, which would greatly strengthen the persuasiveness of the protocol.
- The manuscript mentions that a total of 1530 features were generated, and then an LSTM autoencoder was used for feature selection/dimensionality reduction, but the description is too general. It is suggested to present the core features selected by the autoencoder to improve clinical interpretability.
- In Section 2.2.1, multiple data augmentation methods (Gaussian noise, time warping, amplitude scaling) are mentioned, and then Section 2.2.2 focuses on Mixup. Please clarify whether these methods are applied sequentially or selectively. Is Mixup applied after the other augmentation methods?
- To ensure reproducibility of the study, please provide additional details on the following experimental aspects: IMU calibration methods (e.g., static calibration, dynamic calibration); the specific fixation method and anatomical placement of the sensors; and how potential sensor displacement during long-duration or repeated movements was addressed.
- Although the manuscript cites various sensors and algorithms in the literature, it does not provide quantitative comparisons with existing published models. It would be valuable to compare with other studies (e.g., the multi-sensor approach in [24]) to highlight the advantages of this method.
- The results show good performance metrics (R² > 0.82). Considering the small sample size, it is recommended to provide confidence intervals or perform appropriate statistical tests to enhance the statistical rigor of the results.
- The future work section is well written. It could further mention the potential integration of this system with remote rehabilitation platforms, which would better demonstrate the long-term value of this technology.
Reviewer 2 Report
Comments and Suggestions for Authors
This manuscript presents a deep learning–based framework using IMU sensor data to estimate Fugl-Meyer Assessment Upper Extremity (FMA-UE) subdivision scores. The topic is timely and relevant to remote stroke rehabilitation, and the study demonstrates promising results. However, several major issues should be addressed before the work can be considered for publication.
- The study includes only 15 participants with an imbalanced distribution of healthy and stroke subjects. The small sample size limits the generalizability of the findings. Is the data obtained from a small number of sample tests representative?
- While IMU data collection and preprocessing are described, critical details regarding feature selection, network architecture, and hyperparameter settings are missing. Please provide a clearer description of these components.
- The output of the IMU sensor is a voltage signal. How did the author convert these voltage signals into feature signals and input them into deep learning? Is there any theoretical basis for the installation of four IMUs on the human arm and trunk? Why install it at these locations?
- Please provide the original test data of some IMUs and give the differential characteristics of the output signals of IMUs at different positions.
Reviewer 3 Report
Comments and Suggestions for Authors
Thank you for the opportunity to review your manuscript Overall, a very valuable study with potentially very good outcomes for delivering post stroke rehabilitation. The study was generally well written. The introduction provided good context with relevant and appropriate references and justified the development of the estimator in practice. The methods were clearly defined and were appropriate for this study. The results were presented well; the visuals were easy to follow and linked in with your text clearly. Some further text to describe the findings would have been useful (see below comments). The discussion would benefit from some further refinement, notably the inclusion of supporting literature. I have made some specific comments below.
Line 67 - replace "takes" with "includes"
Methods section: Should be written in past tense. Sections 2.1.1, 2.1.2 and 2.1.3 will need to be re-phrased.
Line 188 - 204 - There are some issues with wording in this paragraph. For example, Line 188 should read "expected to be...", Line 198 should read "needed to be repeated..." I suggest a thorough review of this section.
Line 381 - remove full stop.
Discussion - I feel that some of your discussion belongs in the results section. For example, Lines 396 - 413. This information is not a discussion about your findings but more of a description (results). It is also not linked with literature or higher-level findings.
Discussion - There is limited reference to supporting literature, which would be beneficial to demonstrate how your findings can be generalized and their applicability to the real-world setting. I suggest looking at integrating more supporting literature and providing a higher level summary.
Conclusion - I would remove any reference to your results in the conclusion and focus on an overall summary.
Comments on the Quality of English LanguageThe language is mostly fine, some minor adjustments would help with the readability. I have noted some specific sentences that could be improved.
Reviewer 4 Report
Comments and Suggestions for Authors
This study presents an LSTM autoencoder with Mixup augmentation to estimate Fugl Meyer Assessment Upper Extremity (FMA-UE) subdivision scores from IMU sensor data collected in stroke survivors. The work is interesting especially in focusing on subdivision scores ( hand, shoulder/elbow, wrist, coordination) rather than total FMA-UE and the results show strong predictive performance (R² > 0.82 across all subdivisions, overall R² about 0.97) with low normalized errors. The methodology is described well: seven targeted motions, LOSOCV evaluation, a 15 subject dataset, standardized sensor placement, and detailed preprocessing steps all support reproducibility. The addition of reduced motion experiments provides additional insight into the importance of motion diversity for estimator accuracy.
However, please imporve the following for the work to be considered for acceptance and publication. The sample size is limited. Please provide more robust support for the results obtained and how you handeled issues such as overfitting, etc. Therefore, in the future work (or limitation section maybe) you should include it as a larger and more diverse patient population (with very mild and severe cases) is needed totest generalizability.
Moreover, can the authors clarify why exactly these seven motions were chosen and whether fewer or alternative motions might achieve similar accuracy? How does the proposed model compare numerically with other IMU based or machine learning approaches for FMA estimation published in recent years? Another point concerns model interpretability, which features or motions contribute most to each sub score prediction? Without this, it is difficult to understand whether the model is capturing clinically meaningful movement patterns. Also, the limitations section should be expanded to include what are the risks of ceiling or floor effects in the dataset and how would the model handle imbalanced distributions of impairment severity?
Thank you.
Round 2
Reviewer 1 Report
Comments and Suggestions for Authors
Thanks for the authors' careful response. I recommend to accept in the present form.
Reviewer 2 Report
Comments and Suggestions for Authors
The authors revised well based on my suggestions.